# Bilateral Biceps Curl Shows Distinct Biceps Brachii and Anterior Deltoid Excitation Comparing Straight vs. EZ Barbell Coupled with Arms Flexion/No-Flexion

**DOI:** 10.3390/jfmk8010013

**Published:** 2023-01-19

**Authors:** Giuseppe Coratella, Gianpaolo Tornatore, Stefano Longo, Fabio Esposito, Emiliano Cè

**Affiliations:** 1Department of Biomedical Sciences for Health, Università Degli Studi di Milano, 20133 Milan, Italy; 2IRCCS Galeazzi Orthopaedic Institute, 20161 Milan, Italy

**Keywords:** elbow flexors, EMG, resistance training, weight training, bodybuilder, strength, root mean square

## Abstract

The present study investigated the excitation of the biceps brachii and anterior deltoid during bilateral biceps curl performed using the straight vs. EZ barbell and with or without flexing the arms. Ten competitive bodybuilders performed bilateral biceps curl in non-exhaustive 6-rep sets using 8-RM in four variations: using the straight barbell flexing (ST_flex_) or not flexing the arms (ST_no-flex_) or the EZ barbell flexing (EZ_flex_) or not flexing the arms (EZ_no-flex_). The ascending and descending phases were separately analyzed using the normalized root mean square (nRMS) collected using surface electro-myography. For the biceps brachii, during the ascending phase, a greater nRMS was observed in ST_no-flex_ vs. EZ_no-flex_ (+1.8%, effect size [ES]: 0.74), in ST_flex_ vs. ST_no-flex_ (+17.7%, ES: 3.93) and in EZ_flex_ vs. EZ_no-flex_ (+20.3%, ES: 5.87). During the descending phase, a greater nRMS was observed in ST_flex_ vs. EZ_flex_ (+3.8%, ES: 1.15), in ST_no-flex_ vs. ST_flex_ (+2.8%, ES: 0.86) and in EZ_no-flex_ vs. EZ_flex_ (+8.1%, ES: 1.81). The anterior deltoid showed distinct excitation based on the arm flexion/no-flexion. A slight advantage in biceps brachii excitation appears when using the straight vs. EZ barbell. Flexing or not flexing the arms seems to uniquely excite the biceps brachii and anterior deltoid. Practitioners should consider including different bilateral biceps barbell curls in their routine to vary the neural and mechanical stimuli.

## 1. Introduction

Resistance training is considered a potent stimulus to increase strength [1] through neural [2] and structural adaptations [3]. In this practice, each exercise has a group of prime movers that are primarily stimulated during the movement [4]. Nevertheless, many exercises can be performed in a multitude of variations and examining how each variation stimulates the targeted muscles differently is essential to plan the resistance training sessions purposely [5,6,7,8,9]. 

Among the exercises performed with intent to stimulate the upper limb muscles, the biceps curl is definitely one of the most common. Basically, a biceps curl consists of an elbow flexion against an external resistance that, when constant, can be a dumbbell, a barbell, a cable or a selectorized load on a gym device [10,11,12,13,14,15]. The biceps curl has its main target in the elbow flexors, i.e., brachialis, brachioradialis and biceps brachii. However, while the first two are single-joint muscles, the biceps brachii can act as a wrist supinator and arm flexor [11,16]. Consequently, performing wrist supination and/or arm flexion, as well as placing the wrist and/or the arm in a given position isometrically, can impact the biceps brachii excitation [11,15,17]. Moreover, the biceps curl can be performed unilaterally, e.g., using a dumbbell or cable alternating the limbs or performing a set with a given limb first, or bilaterally, e.g., using two dumbbells simultaneously or a barbell. In the case of a bilateral biceps curl using a barbell, practitioners have two main options: the straight or the EZ barbell. The straight allows a supinated or pronated handgrip, while the handgrip on the EZ barbell depends on its curvature and will be more neutral compared with a pure supinated or pronated handgrip regardless. 

The literature has previously investigated the excitation of the biceps curl prime movers in a series of direct comparisons. For example, elbow flexion performed at different arm flexion angles showed a greater biceps brachii excitation at the longer muscle, i.e., at increased arm extension [15], as also reported in another study [17]. However, in these studies the arm flexion was isometric but not dynamic, and in the practice the latter is often performed, so the information about is scarce. In line with this, arm flexors such as the anterior deltoid have not been extensively examined thus far. When comparing the straight vs. the EZ barbell, the only previous study that has examined the difference in biceps brachii excitation found no difference [12]. This study also split the exercise into ascending and descending phases showing specific differences between the exercises [12]. The importance of investigating the two phases separately derives from the different acute neural patterns [18,19,20], as well as the muscle damage and subsequent protective effects when the descending phase is accentuated [21,22,23], in addition to long-term neural and structural adaptations [24,25]. Moreover, when investigating muscle excitation in resistance exercises, bodybuilders have shown a unique capacity to perform each repetition with a consistent technique, making them suitable for this kind of study [26]. 

Therefore, the present study aims to investigate the muscle excitation of the biceps brachii and anterior deltoid in different bilateral biceps curl variations, varying the barbell, i.e., straight or EZ, and combining the barbell choice with performing or not performing the arm flexion. To deepen the analysis, both the ascending and descending phase were examined, and the biceps curl variations were performed by competitive bodybuilders. 

## 2. Materials and Methods

### 2.1. Study Design

The present investigation was designed as a cross-over, repeated-measures, within-subject study and was conducted in line with previous studies from our laboratory [5,6,7,8,9]. The participants were involved in five different sessions (Figure 1). In the first session, the participants were familiarized with the technique defined for each variation. In sessions two and three in random order, the 8-RM [8] was determined for the biceps curl using straight or EZ barbells, and for each barbell with or without the arm flexion. In the fourth session, the participants were familiarized with the electrodes’ placement while performing all the biceps curl variations. In the fifth session, the muscles’ maximum excitation was first measured. Then, after a minimum of 30 min of passive recovery, the participants performed a non-exhaustive set for each exercise in a random order, with an inter-set pause of 10 min. Each session was separated by at least three days, and the participants were instructed to avoid any further form of resistance training for the entire duration of the investigation. Overall, the present procedures were chosen to avoid any insurgence of fatigue, acute in the range of minutes, or possible exercise-induced muscle damage in the range of days.

### 2.2. Participants

The present investigation was advertised by the investigators during some regional and national competitions, and to be included in the study, the participants had to compete in regional competitions for a minimum of 5 years. Additionally, they had to be clinically healthy, without any reported history of upper-limb and lower-back muscle injury or neurological or cardiovascular disease in the previous 12 months. To avoid possible confounding factors, the participants competed in the same weight category (Men’s Classic Bodybuilding < 80 kg, <1.70 m), according to the International Federation of Body Building Pro-League. The use of drugs or steroids is continuously monitored by a dedicated authority under its regulations, removing the need for independent assessment in this study. Thereafter, 10 male competitive bodybuilders (age 29.8 ± 3.0 years; body mass 77.9 ± 1.0 kg; stature 1.68 ± 0.01 m; training seniority 10.6 ± 1.8 years) were recruited for the present procedures, in line with previous studies [6,7,8,9]. The participants were asked to abstain from alcohol, caffeine or similar beverages in the 24 h preceding the test. After a full explanation of the aims of the study and the experimental procedures, the participants signed a written informed consent. They were also free to withdraw at any time. The current design was approved by the Ethical Committee of the Università degli Studi di Milano (CE 27/17) and performed following the Declaration of Helsinki (1964 and updates) for studies involving human subjects. The individual in this manuscript has given written informed consent to publish these case details.

### 2.3. Exercises Technique

The biceps curl variations were performed using a straight barbell (Technogym, Cesena, Italy), or an EZ barbell (Technogym, Cesena, Italy). The technique of each exercise is shown in Figure 1 and Figure 2, and each exercise was performed standing. For the straight barbell, the handgrip was supine with an inter-hand distance derived from the arms parallel and the valgus of elbow, thus larger than the shoulder distance (Figure 2A). For the EZ barbell, the handgrip was orientated in a more neutral position following the EZ barbell curvature, with an inter-hand distance of approximately shoulder width, depending on the anthropometrics of each participant (Figure 2B). 

Using either the straight or the EZ barbell, the biceps curl was performed not flexing or flexing the humerus, for a total of four variations. Irrespective of the variation, and following recent updates on appropriate descriptions of the resistance exercise technique [4], (i) the load was fixed as 8-RM, (ii) six repetitions were performed—not to failure—to avoid fatigue, (iii) the action was performed with a full range of movement and (iv) with a tempo of 1-2-1-2 s for the first isometric, the ascending, the second isometric and the descending phase, respectively, so that (v) all dynamic phases were performed, while (vi) using an external focus. In all four variations, the arms were close to the trunk. Notably, using the barbell does not allow for a full elbow extension because the barbell will hit the thighs during the descending phase, so the range of movement was stopped when the barbell touched the thighs (Figure 3A). In the two variations in which the arms were not flexed (straight or EZ barbell), the range of movement was based on the elbow flexion-extension (Figure 3B). In the two variations which were flexed (straight or EZ barbell), this was accompanied by a flexion of the arms (approximately 30°) (Figure 3C). The participants were instructed to avoid sagittal oscillations of the trunk or any movement of the lower limbs and to exaggerate the elevation of the scapulae, and the technique was checked by three operators. A visual feedback was provided to help the participants to follow the tempo for each phase [5,9,27]. The set was repeated in case of disproportionate duration of any phase, as for any disproportionate movement of the trunk or lower limbs. 

### 2.4. 8-RM Protocol

The 8-RM was assessed using the same exercise technique described above, following previous procedures [8]. Briefly, after a standardized warm-up consisting of 3 × 15 repetitions of the biceps curl exercise using three incremental self-selected loads, the 8-RM was determined incrementing the load until the eighth repetition corresponded to failure, defined as the incapacity to perform the ascending phase [28]. Each attempt was separated by at least 3 min of passive recovery. Strong standardized encouragements were provided to the participants to maximally perform each trial.

### 2.5. Maximum Voluntary Isometric Excitation

The maximal voluntary isometric excitation of the biceps brachii and anterior deltoid was measured in random order following the SENIAM (surface electromyography for the non-invasive assessment of muscles) procedures [29]. The electrodes’ (mod H124SG Kendall ARBO; diameter: 10 mm; inter-electrodes distance: 20 mm; Kendall, Donau, Germany) placement was in line with the SENIAM recommendations [29]. The electrodes were equipped with a probe (probe mass: 8.5 g, BTS Inc., Milano, Italy) that permitted the detection and transfer of the surface electromyography (sEMG) signal by wireless modality. The sEMG signal was acquired at 1000 Hz, amplified (gain: 2000, the impedance and the common rejection mode ratio of the equipment are >1015 Ω//0.2 pF and 60/10 Hz 92 dB, respectively) and driven to a wireless electromyographic system (FREEEMG 300, BTS Inc., Milano, Italy) that digitized (1000 Hz) and filtered (filter type: IV-order Butterworth filter, band-pass 10–500 Hz) the raw sEMG signals. The electrodes were placed on the dominant limb. 

The sEMG electrodes for the biceps brachii were placed on the line between the medial acromion and the fossa at 1/3 from the fossa cubit [29]. The participants were then instructed to flex the elbow with the hand supinated against manual resistance [29]. The sEMG electrodes for the anterior deltoid were placed over the mid-belly of the muscle approximately 4 cm below the clavicle [29]. The participants were then instructed to flex the elbow to 90° so that the hand was pointed upwards and asked to make a closed fist with the hand of the flexed arm and to provide maximal force to produce shoulder flexion against manual resistance [8]. Each attempt lasted 5 s, and three attempts were completed for each movement interspersed by 3 min of passive recovery [6,8]. The operators provided strong standardized verbal encouragements. In line with previous procedures, the electrodes were placed on the dominant limb [6,9,27].

To check for the appropriate electrode placements, the innervation zone shifts during movements for each muscle were checked by means of an 8 × 8 semi-disposable high-density electrodes matrix for sEMG detection (GR10MM0808 model, inter-electrode distance of 10 mm, OtBiolettronica Turin, Italy), in line with previous procedures [5]. The sEMG signal was acquired by a multichannel amplifier (EMG-USB model, OtBioelettronica, Turin, Italy; input impedance of >90 MΩ; CMRR of >96 dB; EMG bandwidth of 10–500; gain of 1000×). From the analysis of the sEMG signal, the innervation zone was identified, and the muscle area involved in the innervation zone shift during the exercises was avoided. Thereafter, the high-density electrode matrix was removed and replaced by the rounded electrodes.

### 2.6. Data Analysis

The sEMG signals from both the peak value recorded during the maximum voluntary isometric activation and from the ascending and descending phases of each exercise were analyzed in the time-domain, using a 25-ms mobile window for the computation of the root mean square (RMS). For the maximum voluntary isometric activation, the average of the RMS corresponding to the central 2 s was considered. During each exercise, the RMS was calculated and averaged over the 2 s of the ascending and descending phase. To identify the ascending and the descending phase, the sEMG was synchronized with an integrated camera (VixtaCam 30 Hz, BTS Inc., Milano, Italy) that provided the duration of each phase [8,9,27]. Such a duration was used to mark the start and end of each phase while analyzing the sEMG signal. The sEMG data were averaged excluding the first and the last repetition of each set, to achieve a more consistent technique and decrease the interference of fatigue [30]. Afterward, the sEMG RMS of each muscle during each exercise was normalized (nRMS) for its respective maximum voluntary isometric activation [6,8,9,27] and inserted into the data analysis. 

### 2.7. Statistical Analysis

The statistical analysis was performed using a statistical software (SPSS 22.0, IBM, Armonk, NY, USA). The normality of data was checked using the Shapiro–Wilk test and all distributions were normal (*p* > 0.05). Descriptive statistics (participants = 10) are shown as the mean (SD). The differences in the nRMS were separately calculated for the biceps brachii and anterior deltoid using a barbell (2 levels: straight and EZ) x arm flexion (2 levels: arms flexed and not flexed) x phase (2 levels: ascending and descending phase) repeated-measures ANOVA. Multiple comparisons were adjusted using the Bonferroni’s correction and reported as mean difference with 95% confidence interval (95%CI). Significance was set at α < 0.05. The magnitude of the interactions was calculated using partial eta squared (η_p_^2^) and interpreted as trivial (up to 0.009), small (0.010 to 0.059), medium (0.060 to 0.139) and large (≥0.140) [31]. The pairwise differences are reported as the mean with a 95% confidence interval with Cohen’s d effect size (ES), and ES was interpreted according to Hopkins’ recommendations: 0.00–0.19: trivial; 0.20–0.59: small: 0.60–1.19: moderate; 1.20–1.99: large; ≥2.00: very large [32].

## 3. Results

Figure 4 shows the nRMS found in the biceps brachii. A significant and large barbell x flexion x phase interaction was observed (F_1,9_ = 5.477, *p* = 0.044, η_p_^2^ = 0.378). During the ascending phase, a greater nRMS was observed when the arms were not flexed using the straight compared with the EZ barbell (+1.8%, +0.6% to +2.9%, ES: 0.74). Moreover, a greater nRMS was observed when flexing vs. not flexing the arms using both the straight (+17.7%, +13.3% to +22.2%, ES: 3.93) and the EZ barbell (+20.3%, +16.8% to +23.8%, ES: 5.87). During the descending phase, a greater nRMS was observed when the arms were flexed using the straight compared with the EZ barbell (+3.8%, +0.1% to 7.5%, ES: 1.15). Moreover, a greater nRMS was observed when not flexing vs. flexing the arms using both the straight (+2.8%, +0.2% to 5.4%, ES: 0.86) and the EZ barbell (+8.1%, +4.8% to +11.4%, ES: 1.81). The nRMS was greater (*p* < 0.05) during the ascending than the descending phase whatever the exercise variation. 

Figure 5 shows the nRMS found in the anterior deltoid. A non-significant but large barbell x flexion x phase interaction was observed (F_1,9_ = 4.284, *p* = 0.068, η_p_^2^ = 0.322). However, significant and large barbell x flexion (F_1,9_ = 126.323, *p* < 0.001, η_p_^2^ = 0.990 large) and flexion x phase interactions (F_1,9_ = 711.0.36, *p* < 0.001, η_p_^2^ = 0.988 large) were observed. During the ascending phase, a greater nRMS was observed when the arms were not flexed using the straight compared with the EZ barbell (+5.8%, +3.7% to +7.9%, ES: 2.91) and when the arms were flexed using the EZ compared with the straight barbell (+10.8%, +6.2% to +15.4%, ES: 2.34). During the descending phase, a greater nRMS was observed when the arms were not flexed using the straight compared with the EZ barbell (+2.3%, +0.5% to +4.6%, ES: 0.96) and when the arms were flexed using the EZ compared with the straight barbell (+9.0%, +7.5% to +10.5%, ES: 3.15). The nRMS was greater when flexing compared with not flexing the arms (*p* < 0.05) and during the ascending than the descending phase (*p* < 0.05) whatever the exercise variation. 

## 4. Discussion

The present study was conceived to investigate the excitation of the biceps brachii and anterior deltoid in different biceps barbell curl variations, exploring the effects of different barbells (straight or EZ), and different techniques (flexing or not flexing the arms) during both the ascending and descending phases. The straight barbells induced a moderate increase in biceps brachii excitation compared with the EZ barbell, while flexing the arms induced very large increases during the ascending phase. Interestingly, the biceps brachii was more excited when not flexing vs. flexing the arms during the descending phase. As concerns the anterior deltoid, the straight barbell induced more excitation compared with the EZ barbell when not flexing the arms and—in contrast—less excitation when flexing the arms. The anterior deltoid was more excited when flexing vs. not flexing the arms, and both the biceps brachii and anterior deltoid were more excited during the ascending vs. the descending phase. Accordingly, the overall findings highlighted that both the barbell and the arm flexion stimulate the biceps curl prime movers differently. Many novel aspects of the study design, i.e., the examination of the excitation of the anterior deltoid, the comparison between active arm flexion vs. no flexion and the population recruited are quite novel in the literature; thus, when no comparison could be made, we provided anatomical and biomechanical explanations for the present results. 

The straight vs. EZ barbell allows for a more supinated grip, so as to theoretically favor the biceps brachii which is a wrist supinator. However, such a theoretical superiority was only visible during the ascending phase when not flexing the arms and during the descending phase when flexing the arms, and in both cases to only a moderate extent. A previous study reported mostly equivalent biceps brachii excitation when directly comparing the two barbells, also observing no difference in the brachioradialis excitation [12]. The slight discrepancy between this and the present study may depend on both the load selected (60% 1-RM vs. 8-RM, respectively) and the participants’ backgrounds (3-yrs training experience vs. competitive bodybuilders, respectively). However, while the supination recall for greater biceps brachii excitation while flexing the elbow compared with the other two elbow flexors brachioradialis and brachialis [16], it is possible that the dissimilar design between the straight and the EZ barbell might be not marked enough to highlight systematic differences in biceps brachii excitation when the wrist is supinated or almost supinated. 

While acknowledging that the non-significant barbell x flexion x phase interaction, albeit with a large effect size, may derive from the limited sample size, the anterior deltoid showed an interesting behavior as the factor of flexion appears to produce curious consequences. Indeed, when not flexing the arms, a greater excitation was observed using the straight vs. EZ barbell in both the ascending (ES: very large) and descending (ES: moderate) phases. On the contrary, when flexing the arms, the anterior deltoid was more excited using the EZ vs. straight barbell in both the ascending (ES: very large) and descending (ES: very large) phases. Performing the biceps curl with the handgrip we chose as the technique for the straight barbell, i.e., larger than the shoulders, forces the humerus to be greatly externally rotated compared with the external rotation necessary to grip the EZ barbell. Consequently, the anterior deltoid is much lengthened as it acts as an internal humerus rotator [33]. The EMG signal appears to be sensitive to the muscle elongation, with a greater signal observed at the longer muscle length [34]. Therefore, the greater external rotation of the humerus without any other movement, i.e., when not flexing the arms, may explain the greater excitation of the anterior deltoid using the straight vs. EZ barbell. In contrast, flexing the arms greatly engages the anterior deltoid as the prime mover [33]. The non-excessive external rotation using the EZ barbell may place the anterior deltoid in a more favorable moment arm [33], recreating the “full can” position of the lateral raises [8]. This may explain the greater nRMS recorded during using the EZ vs. straight barbell. 

When flexing the arms, the biceps brachii was more excited during the ascending (ES: very large) and less excited during the descending phase (ES: moderate to large) compared with not flexing the arms. In contrast, the anterior deltoid was markedly more excited when flexing vs. not flexing the arms. During the ascending phase, the barbell must be accelerated by the prime movers, and flexing the arms creates the context to emphasize the work by the arm flexors, i.e., the biceps brachii and anterior deltoid in the case of the present study. It may be argued that during the descending phase, the movement is mainly controlled by an eccentric movement of the elbow flexors when the arms are not flexed, while the inertia of the barbell requires a more collaborative control of the elbow and arm flexors when returning from the biceps curl performed flexing the arms. Nonetheless, the role of other important arm flexors such as the pectoralis major with its clavicular head was not examined, and as such it is not possible to assert what role each arm flexor may have had. Notably, the electrodes’ placement following the SENIAM recommendations did not allow us to distinguish the short and the long head of the biceps brachii [29,35], so the nRMS recorded here must be intended for biceps brachii as a whole, including its role in both elbow and arm flexion. 

The nRMS of both the biceps brachii and anterior deltoid were greater during the ascending compared with the descending phase, whatever the biceps curl variation. The neuromuscular uniqueness of the eccentric vs. concentric contraction [18,19,20], and the additional use of the semi-passive and passive sarcomeres proteins [36,37,38], make the eccentric contraction less expensive, with a lower nRMS recorded for a given load [39]. Therefore, using a constant external load is a sub-stimulus for the eccentric phase, since it is tailored on the concentric capacity [40]. However, the present results show that the biceps curl variations examined may have different neural patterns during the ascending or descending phase, so it would be relevant to plan all of them in resistance training practice. 

Some limitations accompany the present investigation. First, the excitation of the brachioradialis was not recorded and would have provided deeper information about the role of the elbow flexors. However, we followed the SENIAM recommendations, which do not include the brachioradialis among the muscles clearly assessable by sEMG. Second, we did not record the excitation of the pectoralis major, and this would have deepened the aspects concerning the arm flexion. Third, the present results refer to the present population and sample size and the detailed technique we used, and hence they should not be generalized. Last, we are aware that the biceps curl includes many more variations that remain to be examined.

## 5. Conclusions

The present study examined the excitation of the biceps brachii and anterior deltoid in different bilateral biceps barbell curl variations. While the straight barbell appears slightly advantageous compared with the EZ barbell as concerns the biceps brachii excitation, the anterior deltoid was more excited using the straight barbell when not flexing the arms, and more using the EZ barbell when flexing the arms. The arm flexion induced a greater excitation of the anterior deltoid, while the biceps brachii was more excited during the ascending phase, and less excited during the descending phase. Lastly, the ascending phase excited the biceps brachii and anterior deltoid more than the descending phase.

In training practice, inserting different bilateral biceps barbell curl variations may be considered to stimulate the prime movers differently. The combination of different barbells and flexing/not flexing the arms can be used to vary the neuromuscular and mechanical stimuli to the prime movers and to stimulate the prime movers in different ways. Moreover, the ascending and descending phases appear to have different neural schemes and may be performed together, or even separately with a tailored load for each one.

## Figures and Tables

**Figure 1 jfmk-08-00013-f001:**
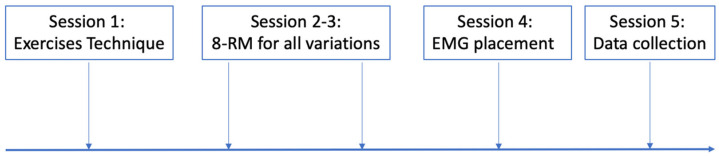
The study design.

**Figure 2 jfmk-08-00013-f002:**
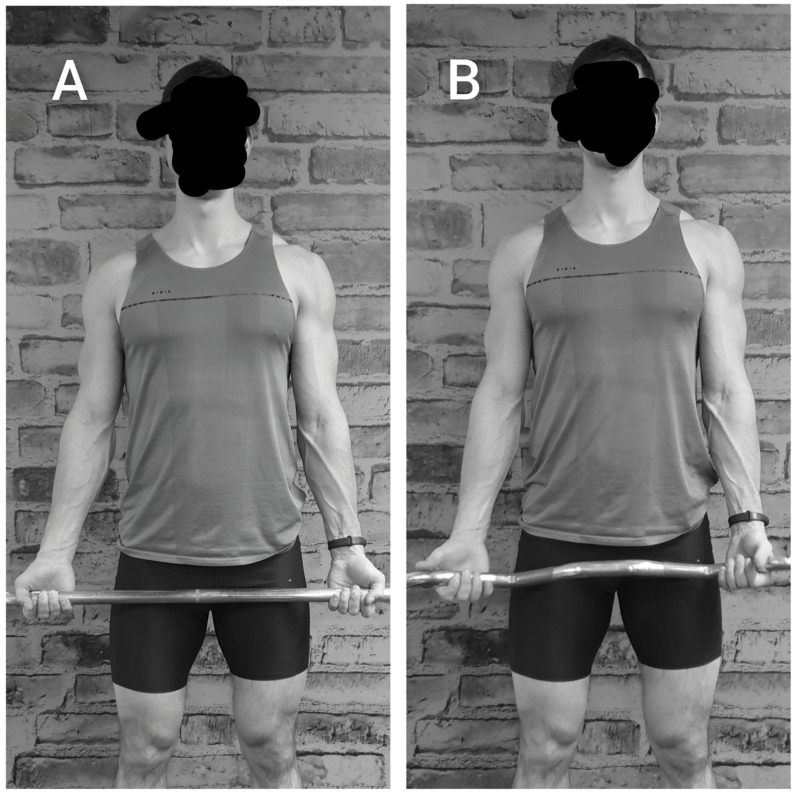
The grip used for the straight (**A**) and the EZ barbell (**B**).

**Figure 3 jfmk-08-00013-f003:**
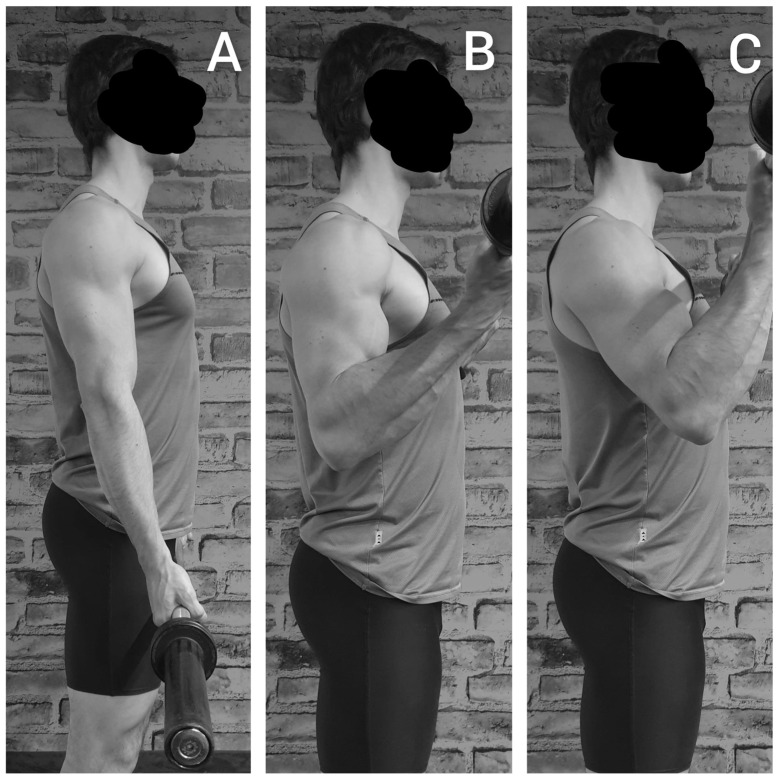
For both the straight and the EZ barbell, the start of the movement (**A**) and the end of the movement without (**B**) or with arm flexion (**C**).

**Figure 4 jfmk-08-00013-f004:**
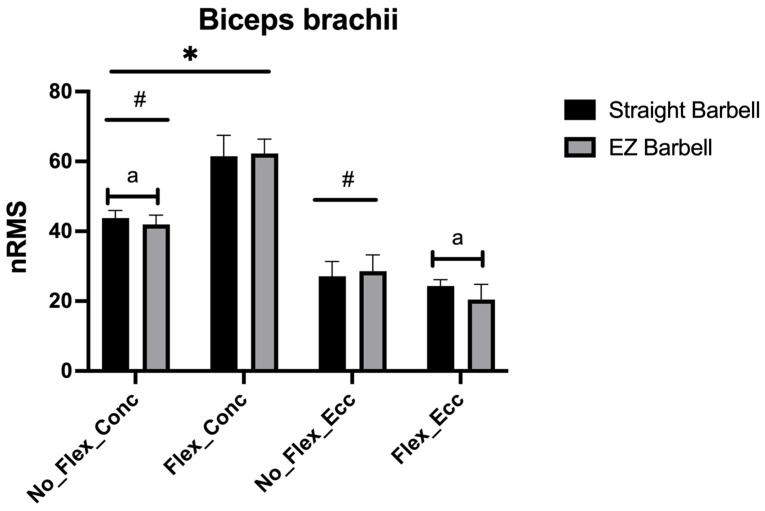
The nRMS for the biceps brachii is shown. a: *p* < 0.05 vs. the EZ barbell; #: *p* < 0.05 vs. flexing the arms; *: *p* < 0.05 vs. the descending phase.

**Figure 5 jfmk-08-00013-f005:**
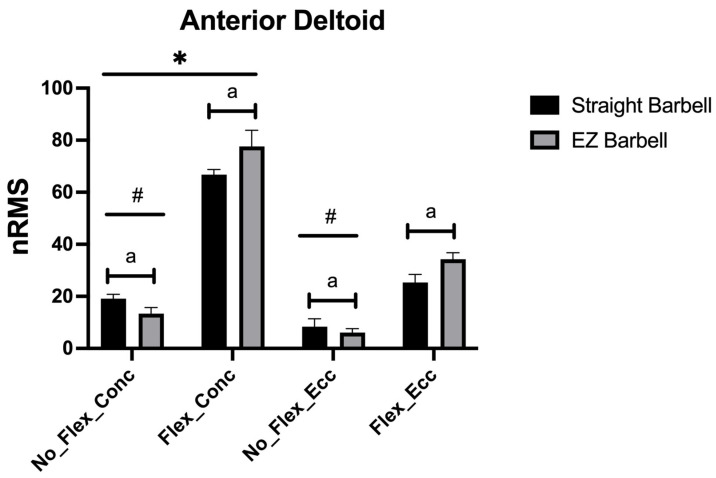
The nRMS for the anterior deltoid is shown. a: *p* < 0.05 vs. the EZ barbell; #: *p* < 0.05 vs. flexing the arms; *: *p* < 0.05 vs. the descending phase.

## Data Availability

Data are available on request to the corresponding author.

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
