# Peer review of "Bilateral Biceps Curl Shows Distinct Biceps Brachii and Anterior Deltoid Excitation Comparing Straight vs. EZ Barbell Coupled with Arms Flexion/No-Flexion"

_jfmk, 2023, doi:10.3390/jfmk8010013_

Round 1

Reviewer 1 Report

Dear Authors,

I hope you are doing very well.

I can understand the pertinence and relevance of this work, however there are issues on methods and discussion that need to be revised. Mainly, at discussion section you need to contrast your findings with previous work and literally 'discuss' them. Also, you citied many of your previous works (self-citing), what led me to question: is your research group the only one investigating these questions in you research field? I am asking genuinely, not provoking. I do not even investigate in this area. Anyway, I believe that this is also something to revise carefully. 

I hope the comments attached can help you improving your work

Kind regards

Author Response

Reviewer 1

I hope you are doing very well.I can understand the pertinence and relevance of this work, however there are issues on methods and discussion that need to be revised. Mainly, at discussion section you need to contrast your findings with previous work and literally 'discuss' them.

Response: We thank the reviewer for the comments. As specific for this comment, the procedures used for this study (i.e., the exercises, the population) have not been used before. Therefore, finding previous works to discuss with is actually challenging. We have done our best to retrieve similar previous results, but here we could only justify what happened without a direct comparison with past studies.

Also, you cited many of your previous works (self-citing), what led me to question: is your research group the only one investigating these questions in you research field? I am asking genuinely, not provoking. I do not even investigate in this area. Anyway, I believe that this is also something to revise carefully. I hope the comments attached can help you improving your work. Kind regards.

Response: We understand your perspective. However, the self-citations derive mainly from methodological points, since this is the sixth of a series of manuscripts in which we investigated muscle excitation in bodybuilders (the real novelty) performing different variations of many exercises. Therefore, we just want to underline that the procedures have been already used. We hope the reviewer could understand.

#1 – In this section is missing information about how data were collected

Response: Added as suggested, please see text.

#2 – “Practitioners should consider including different bilateral biceps barbell curl in their routine”, why? This info/explanation is missing. You can use some of the insights shared at the conclusion section If you need more space to add this information, you can remove the numbers when you describe the results 25-26

Response: Added as suggested, please see text.

Introduction

#3 – the second part of this sentence is very confused. Please, clarify it 33-35

Response: Clarified as suggested, please see text.

#4 – I believe that the writing of this ‘problematization’ can be more readable. Can you revise it, please? 60

Response: We have accepted your suggestion in the following comment. Please see text.

#5 – “In line, the arm flexion like the anterior deltoid were not extensively examined in this research field so far.” I suggest this type of writing 60-61

Response: Changed as suggested, please see text.

Materials and Methods

#6 – Many explanations are needed here: why 30min of passive recovery? In detail, why 30min, why not 35? And why passive and not active recovery? Then, why 10min of pause, why not 13min? Why 3days, why not 2 or 4? 86-90

Response: The overall intent of these procedures was to avoid any insurgence of fatigue, acute in term of minutes, or possible exercise-induced muscle damage when dealing with days. There is not a specific reason for choosing 30 min and not 29 or 31, as well as for the number of days, provided that the previous conditions were respected. As a whole, these procedures have been used in our previous studies (tailored for each specific study design), and that’s why we believe it is important to refer to those studies, as now done at the beginning of the section. We hope we have clarified this issue.

#7 – This sentence is too long, and thereby very confused. Can you please reword it? 126- 130

Response: Actually, these are the variables that describe the exercises variations. We have now numbered as a list, so that the intent is clear. Please see text.

#8 – A dot is missing here 188

Response: added.

#9 – “Descriptive statistics of the selected participants are reported as mean (M) and standard deviations (SD)”. I suggest changing accordingly. Moreover, where is the table with these descriptive values? I know that you present part of them in the figure, but a table with the exact numbers is needed. 212- 213

Response: Actually, we believe that graphs are much more communicative than tables in this kind of study and reporting the same results in a table will be redundant. Additionally, the exact values would not provide anything “crucial”. Therefore, we have changed “reported” in “shown”. Please see text.

#10 – You must interpret the partial eta-square as well. This is crucial (please, consider this comment also for the results section) 218- 219

Response: added as suggested, please see text.

Results

#11 – “95% confidence interval”, this info is not needed, once you mentioned that before at the statistical analysis subsection 227

Response: Deleted as suggested, please see text.

#12 – Please, interpret the partial eta square (see comment 10)

Response: added as suggested, please see text.

Discussion

#13 – I suggest rewording as: “Accordingly, the overall findings highlighted that …” 270- 272

Response: Changed as suggested, please see text.

#14 – All this paragraph is very descriptive. Your findings are not contrasted with previous investigations. Please, revise it accordingly. 289- 305 #15 – the same happens in this paragraph, and in the next on (see previous comment); Therefore, this is a general issue in the discussion section. This is, by far, the section that need more work.

Response: As mentioned in the very first comment, and also as the reviewer may have understood from the introduction, there are many novel aspects in this manuscript. This is especially because the factors we have considered (i.e., straight vs EZ barbell, anterior deltoid as an additional muscle investigated, the role of the humerus flexion) have not been explored before in this combination. The only study that has addressed something similar (Marcolin et al., 2018) has been used for the straight vs EZ barbell comparison. Others are not useful for an actual comparison with the literature. Therefore, we believe that explaining ourselves the present results by an anatomical and biomechanical perspective (as done here) is the most we could do. We hope to have clarified this point.

Reviewer 2 Report

Congratulations to the authors for managing to pull up such a splendid work.
Introduction:

The authors managed to explain the importance of the study, however, would suggest the authors to add on more information on how different styles of barbell training and phases of muscle activation affects the overall picture of an athlete. 

Materials and methods: 

This section is concise and very informative. Suggest the authors to describe some of the methods, i.e. the electrode placements suggested by SENIAM, the flow of the study using figures/pictures. This is to make the information more digestible for the readers. 

Discussion: 

Would suggest the authors to correlate the findings with the anatomically with the neural pathway of the human body. 

How did the authors tackle/resolve the few limitations: usage of brachioradialis and pectoralis major/minor in this study? 

Author Response

Reviewer 2

Congratulations to the authors for managing to pull up such a splendid work.

Response: We thank the reviewer and we hope to have addressed properly all the raised concerns.

Introduction:

The authors managed to explain the importance of the study, however, would suggest the authors to add on more information on how different styles of barbell training and phases of muscle activation affects the overall picture of an athlete. 

Response: We understand the point, but correlating the muscle activation and barbell training to performance or hypertrophy is something far from a direct link. This is not because barbell training does not count as a factor, but rather because there are so many factors that simultaneously concur to both performance and hypertrophy that we believe would be quite speculative. This was also highlighted in two previous review (Vigotsky et al., 2018 and 2022). Therefore, we prefer maintaining the introduction as in the present form.

Materials and methods: 

This section is concise and very informative. Suggest the authors to describe some of the methods, i.e. the electrode placements suggested by SENIAM, the flow of the study using figures/pictures. This is to make the information more digestible for the readers. 

Response: While we believe that the electrodes’ placements as in picture would not be as much more representative than the Seniam website, we have created a new figure for the study design.

Discussion: 

Would suggest the authors to correlate the findings with the anatomically with the neural pathway of the human body.

Response:  We believe that the anatomical description was already done. As concerns the neural pathway, it is unclear what the reviewer is pointing. However, the interpretation of the EMG data only should not allow for further explanation. Notwithstanding, we would revise the discussion after a clearer indication from the reviewer.

How did the authors tackle/resolve the few limitations: usage of brachioradialis and pectoralis major/minor in this study?

Response: Actually, we believe we have tried to discuss the results with quite a caution, keeping in mind that we could have only speculated about those muscles. We also believe that future studies need to assess their excitation.

Round 2

Reviewer 1 Report

Dear Authors,

I hope you are doing very well.

Thank you for the work conducted so far. However, there are some points on you work that need to be revised once again.

Kind regards

Material and Methods

Response to comment #6:

 Sorry, but this information is not enough for me. State that you did the same in previous work is not a reason. At least, please add in the text the reasons that you presented to me: “to avoid any insurgence of fatigue, acute in term of minutes, or possible exercise-induced muscle damage when dealing with days”

Response to comments #9 and #10:

Sorry, but the revisions done are not enough. They way as you reported the eta-square is not appropriate (lines 222 and 223) Which are the intervals between values? For instance, 0,01 means small effect? Or just 0,02? Then, and furthermore, the reporting of outcomes is not right as well. You wrote

“ Figure 2 shows the nRMS for found in biceps brachii. Barbell x flexion x phase interaction

was observed (F1,9 = 5.477, p = 0.044, ?p2 = 0.378 large)”, but the report should be done like:

It was observed a large and significant Barbell x flexion x phase interaction (F1,9 = 5.477, p = 0.044, ?p2 = 0.378)

Please, revise all the result section accordingly. And carefully, because in some parts you have non-significant results but large effects, and you must interpret and debate these outcomes in the discussion.

Response to comment #14: I understand the point raised by the authors. In that case, please add this information at the beginning of the discussion.

Author Response

Dear Authors,

I hope you are doing very well. Thank you for the work conducted so far. However, there are some points on you work that need to be revised once again. Kind regards

Response: We thank the reviewer and we hope the present version might reach the desired quality.  

Material and Methods

Response to comment #6: Sorry, but this information is not enough for me. State that you did the same in previous work is not a reason. At least, please add in the text the reasons that you presented to me: “to avoid any insurgence of fatigue, acute in term of minutes, or possible exercise-induced muscle damage when dealing with days”

Response: Clarified as requested, please see text.

Response to comments #9 and #10:Sorry, but the revisions done are not enough. They way as you reported the eta-square is not appropriate (lines 222 and 223) Which are the intervals between values? For instance, 0,01 means small effect? Or just 0,02?

Response: intervals have been added as requested. Please see text.

Then, and furthermore, the reporting of outcomes is not right as well. You wrote“ Figure 2 shows the nRMS for found in biceps brachii. Barbell x flexion x phase interaction was observed (F1,9 = 5.477, p = 0.044, ?p2 = 0.378 large)”, but the report should be done like: It was observed a large and significant Barbell x flexion x phase interaction (F1,9 = 5.477, p = 0.044, ?p2 = 0.378). Please, revise all the result section accordingly. And carefully, because in some parts you have non-significant results but large effects, and you must interpret and debate these outcomes in the discussion.

Response: Changed as requested. Additionally, we have provided further interpretations in the discussion when needed.

Response to comment #14: I understand the point raised by the authors. In that case, please add this information at the beginning of the discussion.

Response: Added as requested, please see text.

Round 3

Reviewer 1 Report

Dear Authors,

Thank you for your work. The paper is ready for acceptance!

Whish you a great 2023!

All the best